

# On the generalizability of artificial neural networks in spin models

**Hon Man Yau[1] and Nan Su[2⋆]**

**1** Institute of Geology and Geophysics, Chinese Academy of Sciences, 100029 Beijing, China
**2** Frankfurt Institute for Advanced Studies, 60438 Frankfurt am Main, Germany

⋆ nansu@fias.uni-frankfurt.de

## Abstract

The applicability of artificial neural networks (ANNs) is typically limited to the models they are trained with and little is known about their generalizability, which is a pressing issue in the practical application of trained ANNs to unseen problems. Here, by using the task of identifying phase transitions in spin models, we establish a systematic generalizability such that simple ANNs trained with the two-dimensional ferromagnetic Ising model can be applied to the ferromagnetic $q$-state Potts model in different dimensions for $q \geq 2$. The same scheme can be applied to the highly nontrivial antiferromagnetic $q$-state Potts model. We demonstrate that similar results can be obtained by reducing the exponentially large state space spanned by the training data to one that comprises only three representative configurations artificially constructed through symmetry considerations. We expect our findings to simplify and accelerate the development of machine learning-assisted tasks in spin-model related disciplines in physics and materials science.

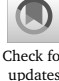

# 1   Introduction

Applications of machine learning techniques in scientific research have flourished in recent years due to advances in both hardware and software, and their ability to identify patterns in data at scale introduced a new paradigm for scientific discovery [1]. This paradigm has been particularly relevant to the computational studies of spin models, notably in condensed matter and statistical physics, materials science and engineering, as they often involve a large amount of data. Supervised learning with ANNs has arguably popularized the use of machine learning algorithms in the study of phase transition [2, 3], and has successfully been applied to various disciplines in physics and materials science [4, 5]. This approach requires that the phase transition of interest is already known so that training data can be correctly labeled.

Due to the lack of a theoretical understanding of the algorithms, data used to test the trained ANNs, and therefore any potential generalizability, is conventionally limited to spin configurations sampled from models that are closely related – for example, ANNs trained with spin configurations of the square-lattice Ising model can also generalize to spin configurations of its triangular counterpart [2,6]. The limited generalizability, due to which computational resources are over-consumed in potentially redundant trainings, severely constrains the general application of machine learning algorithms to unseen problems, and is a pending challenge of the field [7–9].

We propose herein a method that allows us to use ANNs that have only been trained with spin configurations of the simple two-dimensional ferromagnetic Ising model on a square lattice [10, 11] in the classification of spin configurations of the considerably more complex

$Z(q)$-symmetric ferromagnetic Potts models with $q \geq 3$ in different dimensions [12]. This novel generalizability is different than the ANN generalizability to different lattice geometries within the same model or symmetry reported in Refs. [2,6], as it significantly enlarges the applicability of the trained ANNs to unseen, nontrivial problems, especially in the study of phases and critical phenomena of matter and materials described by the Potts model [13]. It is also in contrast to the one reported in Ref. [14] as the two studies focus on different aspects: while Ref. [14] explores the ANN generalizability for frustrated spin models, ours tackles at the level of systematics the ANN generalizability for non-frustrated systems.

Our results show that the trained ANNs are able to use any two out of the $q$ states in a $q$-state Potts model to reproduce critical properties that are generally in good agreement with known values. Furthermore, We demonstrate that the same ideas can be applied to the antiferromagnetic $q$-state Potts model on a square lattice, which is significantly more complex than its ferromagnetic counterpart.

Finally, we explore the impact of reducing the exponentially large state space of the Monte Carlo-sampled spin configurations used to train the ANNs to one that comprises only three representative spin configurations artificially constructed by symmetry considerations, where estimated critical properties also agree well with known values.

## 2 Results and discussions

### 2.1 A generalization scheme of $Z(2)$ symmetry to $Z(q)$ symmetry

#### 2.1.1 Ising model: preparation of the ANNs

We begin by performing supervised learning on fully-connected feed-forward ANNs, which consist of a single hidden layer of 16 neurons and 2 output neurons for carrying out the binary classification of spin configurations. We train the ANNs using spin configurations sampled by Monte Carlo simulations of the two-dimensional ferromagnetic Ising model on a square lattice with nearest-neighbor interactions and periodic boundary conditions, where $\mathcal{H} = -J \sum_{\langle i,j \rangle} \sigma_i \sigma_j$ and $\sigma \in \{-1, 1\}$, and the coupling strength $J$ is set to unity. The trained ANNs are able to classify square-lattice spin configurations of the 2-state ferromagnetic Potts model (Fig. 1a), which is equivalent to the Ising model, in much the same way as reported in the literature [2,6].

We take the intersection of the curves in Fig. 1a as the scale-invariant point with temperature $T'$, and perform finite-size scaling according to the ansatz $\mathcal{W} \sim \widetilde{\mathcal{W}}(tL^{1/\nu'})$, where $t = T - T'$ is the difference between a given temperature $T$ and the estimated critical temperature $T'$, $L$ is the size of the lattice, $\mathcal{W}$ is the output of the ANN, and $\nu'$ is the estimated critical exponent. Finite-size scaling (Fig. 1d) gives an estimate of $T' = 1.1346(3)$ and $\nu' = 0.98(4)$, which agree well with the exact results of $T_c = [\ln(1 + \sqrt{2})]^{-1} \approx 1.1346$ and $\nu = 1$ [13].

These ANNs, which have only processed square-lattice Ising spin configurations during training, are also able to classify spin configurations of the 2-state Potts model on a triangular (Fig. 7a in Appendix C) or honeycomb (Fig. 8a in Appendix C) lattice; and finite-size scaling (Figs. 18a and 19a in Appendix D) gives estimates of $T'$ and $\nu'$ that agree well with the known values (see Table 1 in Appendix B for comparisons). The generalizability of ANNs trained with square-lattice Ising model spin configurations to spin configurations of its triangular-lattice counterpart has previously been noted as a consequence of the models belonging to the same universality class [6], our results for the honeycomb lattice corroborate this explanation.

### 2.1.2 Ferromagnetic Potts model: a transformation enabling generalizability

The Potts model is a generalization of the Ising model from two to $q$ states, which describes a much richer spectrum of phenomena as a result of the enlarged center symmetry $Z(N)$ [13]; its Hamiltonian reads $\mathcal{H} = -J \sum_{\langle i,j \rangle} \delta_{Kr}(\sigma_i, \sigma_j)$, where $\sigma \in \{1, ..., q\}$, and the Kronecker delta $\delta_{Kr}$ evaluates to 1 if $\sigma_i = \sigma_j$ and 0 otherwise. As such, there exists a mismatch in the number of states if one were to feed spin configurations of the Potts model with $q \geq 3$ to ANNs trained only with Ising model spin configurations. To circumvent this mismatch, we map two arbitrarily chosen states to -1 and 1, and treat the rest as having trivial contributions to $\mathcal{W}$ — that is, we introduce sparsity into Potts model spin configurations by applying the following transformation before feeding them to our ANNs, where -1 and 1 simply represent any two of the $q$ states:

$$\{1, 2, 3, ..., q\} \mapsto \{-1, 1, 0, ..., 0\}. \tag{1}$$

After verifying that a configuration consisting entirely of zeros does indeed correspond to a value of $\mathcal{W}$ that is effectively 0, we apply the transformation to spin configurations of the square-lattice Potts model with $q = 3$ and feed them to our ANNs. The maximum value of $\mathcal{W}$ is now approximately 2/3 as a result of removing the contribution of one of the three states; $q/2$-normalized curves are shown in Fig. 1b. The curves intersect at a common point in the same manner as described above for the Ising model, and finite-size scaling (Fig. 1e) gives the estimates $T' = 0.99461(13)$ and $\nu' = 0.85(3)$ (exact: $T_c = [\ln(1 + \sqrt{3})]^{-1} \approx 0.99497$, $\nu = 5/6 \approx 0.833$ [13]).

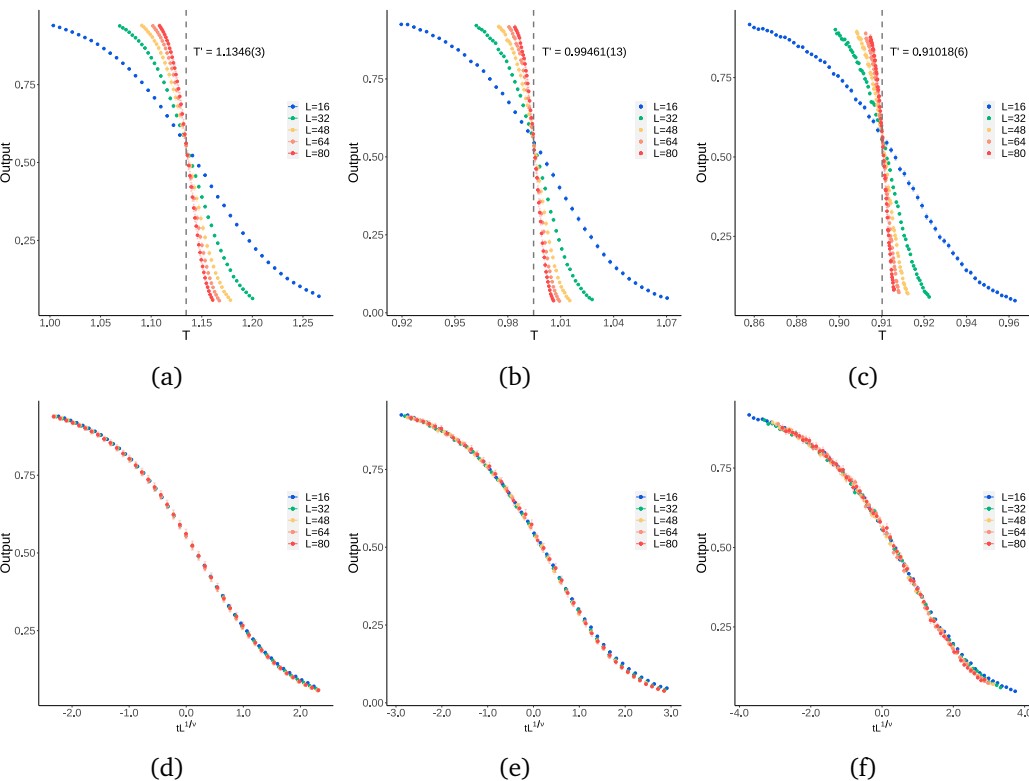

Figure 1: **Generalization of Ising model-trained ANNs to the Potts model. a–c**, Average $q/2$-normalized outputs of Ising model-trained ANNs versus sampling temperature for spin configurations of the Potts models with $q = 2$ (**a**), $q = 3$ (**b**) and $q = 4$ (**c**). **d–f**, Finite-size scaled results of **a–c**. The error bars shown are 99.7% confidence intervals.

Similarly, feeding the same ANNs spin configurations of the square-lattice Potts model with $q = 4$ leads to the curves shown in Fig. 1c, with estimates of $T' = 0.91018(6)$ (exact: $T_c = [\ln(1 + \sqrt{4})]^{-1} \approx 0.91024$ [13]) and, without using any correction terms in finite-size scaling (Fig. 1f), $\nu' = 0.75(2)$, which agrees well with existing numerical studies (Monte Carlo: $\nu \approx 0.722$ [15]). It is well known that correction terms are significant in the finite-size scaling of observables of the 4-state Potts model [16], which is beyond the scope of the current study; however, naively applying the multiplicative logarithmic correction of $(\log L)^{-\frac{3}{4}}$ [15] to finite-size scaling gives an estimate of $\nu' = 0.63(1)$ that is close to the exact value (exact: $\nu = \frac{2}{3} \approx 0.667$ [13]).

We are also able to obtain estimates of critical temperatures that are in good agreement with the general formula of $T_c = [\ln(1 + \sqrt{q})]^{-1}$ [13] for the first-order phase transitions of square-lattice Potts models with $q \geq 5$ (Figs. 6d, 6e, and 6f in Appendix C; see Appendix A.4 for a description of how the corresponding critical parameters are determined). Feeding the ANNs the corresponding triangular- and honeycomb-lattice Potts model spin configurations also give estimates $T'$ and $\nu'$ that are in good agreement with the known values (Figs. 7 and 8 in Appendix C, and Table 1 in Appendix B). These results clearly demonstrate that the generalizability of the ANNs, despite having processed only square-lattice Ising model spin configurations during training, can be extended to the two-dimensional $q$-state Potts model with $q \geq 3$ through a simple transformation of spin configurations according to Eq. (1).

Given that the output of the ANNs is able to correctly describe the physics of the two-dimensional Potts model with different values of $q$ and lattice geometries, we further explore the limit of this generalizability by feeding the ANNs spin configurations of the Potts model in other dimensions. The one-dimensional Potts model does not exhibit a phase transition for any value of $q$ [17], which is consistent with our results that the ANNs do not produce curves with a point of crossing for spin configurations of the one-dimensional Potts model with $q = 2$, 3, and 4 as shown in Fig. 9 in Appendix C.

In the case of the three-dimensional $q$-state Potts model, the estimated values of $T'$ are consistent with Monte Carlo results in the literature, where $T' = 2.2126(14)$ for $q = 2$ (Fig. 2a, Monte Carlo: $T_c = 2.2558$ [18]), $T' = 1.8164(99)$ for $q = 3$ (Fig. 10b in Appendix C, Monte

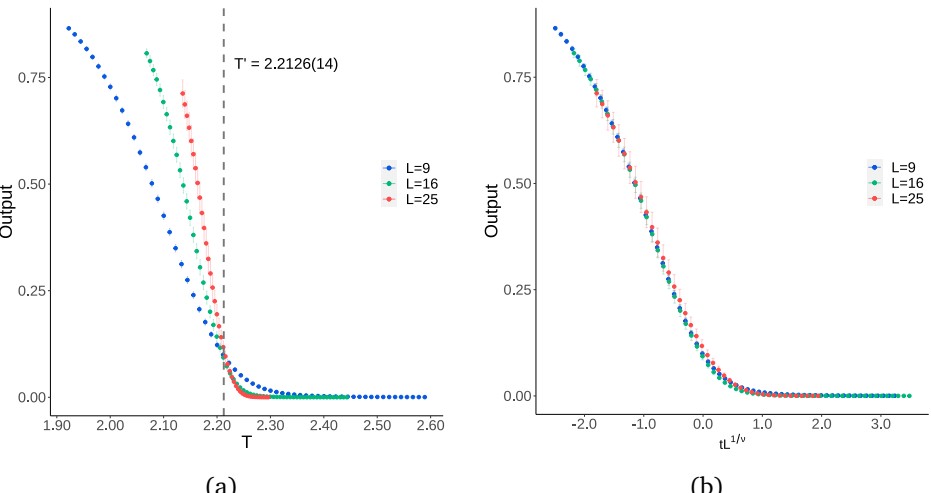

(a)                                                        (b)

Figure 2: **Generalization of two-dimensional Ising model-trained ANNs to the three-dimensional 2-state Potts model. a**, Average outputs of two-dimensional Ising model-trained ANNs versus sampling temperature for spin configurations of the three-dimensional 2-state Potts model. **b**, The associated finite-size scaled results. The error bars shown are 99.7% confidence intervals.

Carlo: $T_c = 1.8163$ [19]), and $T' = 1.5924(15)$ for $q = 4$ (Fig. 10c in Appendix C, Monte Carlo: $T_c = 1.5908$ [20]). In the case of $q = 2$, where the phase transition is second order, the curves collapse upon finite-size scaling as shown in Fig. 2b, giving an estimate of $\nu' = 1.02(2)$, which differs from the known value of $\nu = 0.630$ [18].

We see from these results that the ANNs, which have only learned from spin configurations of the two-dimensional Ising model, are also able to produce an output that allows us to identify the correct critical behaviors when classifying one- and three-dimensional Potts model spin configurations. The exponent $\nu'$ we obtained for the only case that exhibits a second-order phase transition of the three-dimensional 2-state Potts model differs from the known value; however, given that we do observe finite-size scaling behavior, and that the values of $\nu'$ in other two-dimensional lattices remain unaffected, we conjecture that the exponent obtained is a result of the training process encoding the dimensionality of the training data into the ANNs, which is discussed further in Sec. 2.2.

### 2.1.3 Antiferromagnetic Potts model: a nontrivial exploration

The antiferromagnetic Potts model is known to exhibit physics that is significantly richer than its ferromagnetic counterpart [13, 17, 21], and feeding spin configurations of the antiferromagnetic square-lattice Ising model to the ANNs trained with spin configurations of the ferromagnetic square-lattice Ising model always leads to a value of $\mathcal{W} \approx 0$. We train a new set of ANNs with spin configurations of the antiferromagnetic square-lattice Ising model with $\mathcal{H} = J \sum_{\langle i,j \rangle} \sigma_i \sigma_j$, which produce an output that is effectively zero for spin configurations of the ferromagnetic Ising model at all temperatures; in other words, the ANNs are able to distinguish between spin configurations sampled from the ferromagnetic and antiferromagnetic Ising Hamiltonians.

We use the newly trained ANNs to perform classification on spin configurations of the antiferromagnetic Potts model, where $\mathcal{H} = J \sum_{\langle i,j \rangle} \delta_{Kr}(\sigma_i, \sigma_j)$, using the same transformation described in Eq. (1). For $q = 2$, where the exact values of $T_c$ and $\nu$ are the same as the ferromagnetic model, we obtain results that are effectively the same as the ferromagnetic case (Fig. 3a), with values of $T' = 1.1344(3)$ and $\nu' = 1.00(1)$ obtained from finite-size scaling (Fig. 3b). The antiferromagnetic Potts model with $q = 3$ on a square lattice has a highly degenerate ground state and is predicted to be critical at $T = 0$ [22]; our result of a maximum output of $\mathcal{W} \approx 0.08$ near $T = 0$ and the lack of a crossing point as shown in Fig. 3c are consistent with those known features of the model [23, 24]. The output $\mathcal{W}$ for $q = 4$ simply remains at the baseline (Fig. 3d), as is consistent with the prediction that the model is disordered at $T \geq 0$ [24].

Considering the fact that both the ground states of the 3- and 4-state antiferromagnetic Potts models are highly degenerate in a finite volume (infinitely degenerate in the continuum), these results demonstrate the ability of the ANNs, which have only processed spin configurations of the antiferromagnetic square-lattice Ising model with a two-fold degenerate ground state, in detecting orderedness and correctly describing the physics of these highly nontrivial and unseen models.

## 2.2 Exponential reduction of training-data state space

We notice in the experiments described above that spin configurations in the disordered phase sampled at $T \gg T_c$ lead to an output of $\mathcal{W} \approx 0$, and the output of a spin configuration that consists entirely of zeros mirrors this outcome. Inspired by these observations, we train ANNs with only three artificial spin configurations constructed based on symmetry considerations: the $Z(2)$-degenerate spin configurations, $\{1, 1, \ldots, 1\}$ and $\{-1, -1, \ldots, -1\}$, that represent the ordered phase; and the spin configuration $\{0, 0, \ldots, 0\}$ that represents the disordered phase.

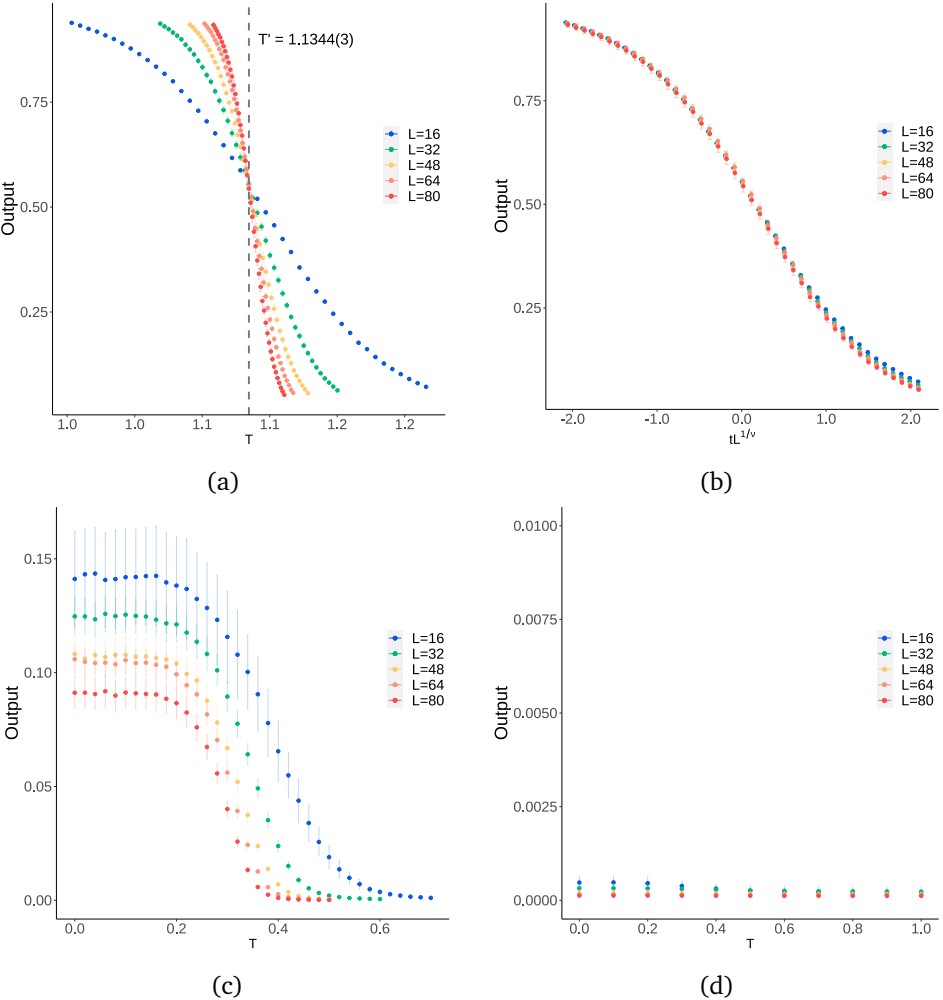

Figure 3: **Application of our methodology to the antiferromagnetic Potts model.**
**a–b**, Average $q/2$-normalized outputs of antiferromagnetic Ising model-trained ANNs
versus sampling temperature for spin configurations of the antiferromagnetic Potts
models with $q = 2$ (**a**) and the associated finite-size scaled results (**b**). **c–d**, The
corresponding results for $q = 3$ (**c**) and $q = 4$ (**d**). The error bars shown are 99.7%
confidence intervals.

In this way, the $\mathcal{O}(2^{L \times L})$ state space of the Monte Carlo-sampled data used to train the ANNs
described above gets exponentially reduced to an artificial one that is $\mathcal{O}(1)$.

The outputs of these ANNs produce the curves shown in Fig. 4. We see that when presented
with spin configurations of the 2-state Potts model, they are able to produce outputs that
give values of $T' = 1.1364(12)$ and $v' = 1.01(11)$ with finite-size scaling, which are in good
agreement with the corresponding exact values. Using the transformation described in Eq. (1),
we also obtained estimates of $T'$ and $v'$ for all values of $q$ and two-dimensional geometries
that are similar to those described above for ANNs trained with Monte Carlo-sampled spin
configurations (Table 3 in Appendix B). We note that the curves are shaped differently to,
and the errors of outputs in the critical region here are larger than, the corresponding outputs
from ANNs trained with Monte Carlo-sampled spin configurations, which we attribute to much
greater degrees of freedom in weights and biases of the ANNs during training caused by a lack
of information in that region. However, it is significant to note that such a simple setup allows
us to estimate critical properties of the $q$-state Potts model to this level of accuracy.

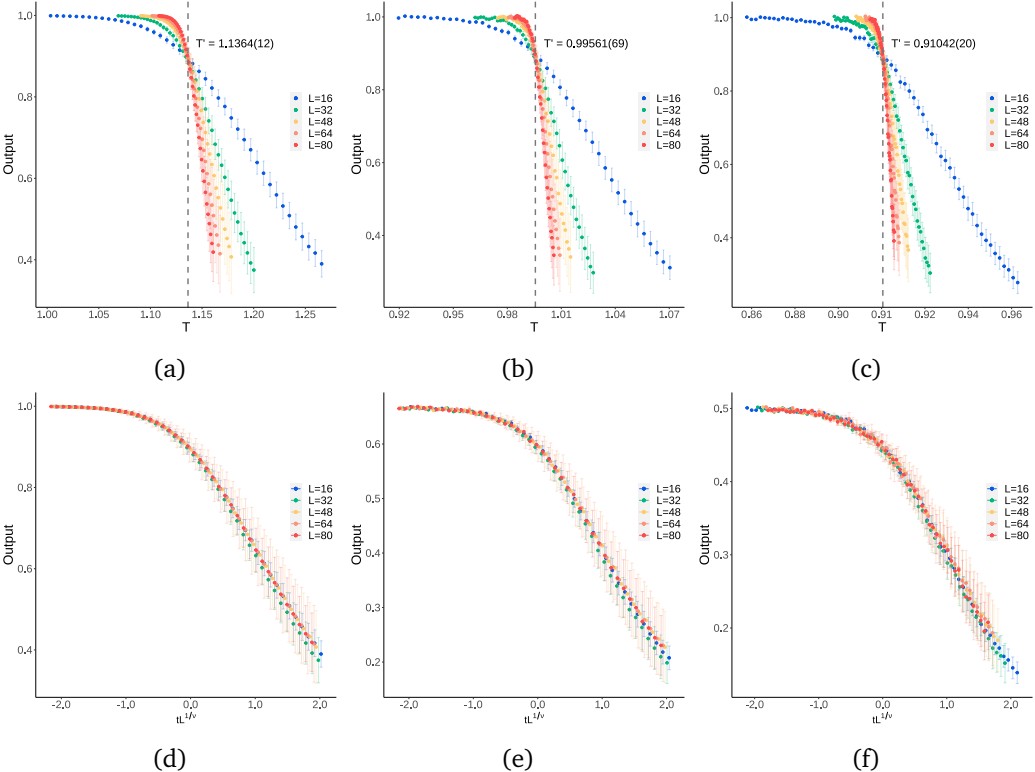

Figure 4: **Exponential reduction of the state space of training data. a–c**, Average $q/2$-normalized outputs of ANNs trained with the reduced state space versus sampling temperature for spin configurations of the Potts models with $q = 2$ (**a**), $q = 3$ (**b**) and $q = 4$ (**c**). **d–f**, Finite-size scaled results of **a–c**. The error bars shown are 99.7% confidence intervals.

Once again, in one dimension we do not observe a crossing point in the output curves (see Fig. 15 in Appendix C), which is consistent with the fact that there is no finite-temperature phase transition for any value of $q$. In three dimensions, the values $T' = 2.2502(31)$ and $\nu' = 0.68(31)$ of the 2-state model (Fig. 5) are noticeably closer to the known values than those obtained using the ANNs trained with Monte Carlo-sampled spin configurations. This observation and the fact that the three representative spin configurations employed in the training are constructed independent of dimensionality support our aforementioned conjecture that the dimensionality of the training data is encoded into the ANNs during training. There is little difference in the values of $T'$ for the three-dimensional Potts model with $q > 2$, which exhibits first-order phase transitions (Table 3 in Appendix B).

The ANNs trained with data from the exponentially reduced state space, which comprises only the three artificial spin configurations chosen based on symmetry considerations, clearly accommodates enough of the underlying physics of the $q$-state Potts model to give outputs that lead to estimates of critical properties that agree well with the known values. This novel reduction may help to improve the interpretability and explainability of machine learning algorithms used in the study of spin models.

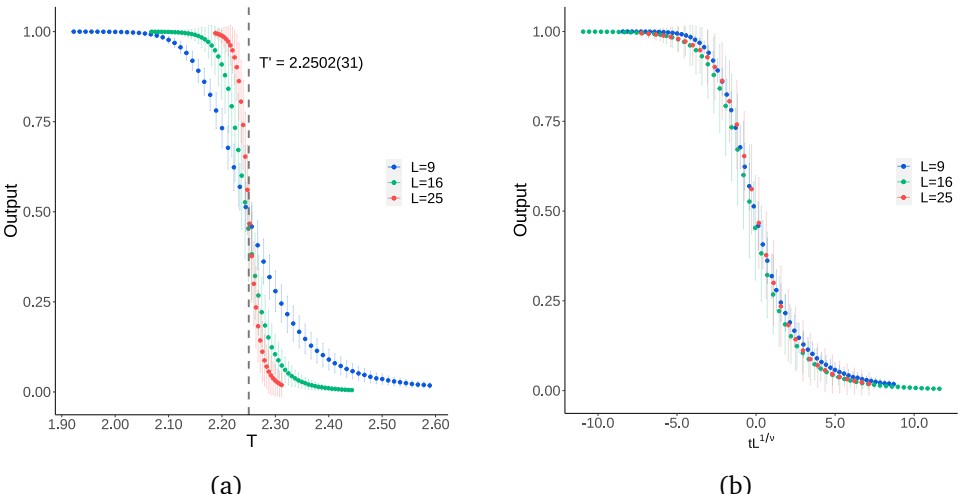

Figure 5: **Three-dimensional test of the state-space reduction of training data.**
**a**, Average outputs of ANNs trained with the reduced state space versus sampling
temperature for spin configurations of the three-dimensional 2-state Potts models.
**b**, The associated finite-size scaled results. The error bars shown are 99.7% confidence intervals.

## 3 Conclusions

By introducing sparsity into spin configurations of the $q$-state Potts model, we have for the
first time uncovered a type of generalizability across symmetries and dimensions for ANNs
that have only been trained with spin configurations of the two-dimensional Ising model. Our
results show that nontrivial information such as critical properties of multiple-state systems
can be encoded in a representation of far fewer states: in this prototypical study, the critical
properties of the $q$-state Potts model are encoded in any two of the $q$ states. This generalizability is conventionally unreachable by performing supervised learning on data drawn from a
single spin model, which – considering the broad applications of the Potts and related models
in various areas [13] – greatly simplifies the machine learning tasks in a spectrum of problems similar to this study, especially in the study of phases and critical phenomena of matter
and materials. As a consequence, shorter development times and more optimized computing
resources in the development of machine learning models can be achieved by reducing the
redundant number of ANNs required in the training.

In addition, we have shown that similar results can be obtained using ANNs trained with
data belonging to a set of only three artificial spin configurations. This reduction of an exponentially large state space of the training data to one that is trivial in size was achieved
systematically, which has the potential to be used as a tool for developing a theoretical understanding of ANNs used in the study of spin models. An analysis of the ANN structures may
reveal deeper insights about why such a minimal training strategy works, and this reduction
may introduce simplification to similar tasks such as quantum machine learning. We plan to
pursue these in a future work. We envisage that the methodologies introduced here will find
general utility in advancing the understanding and development of machine learning techniques applied to various disciplines such as condensed matter physics, high energy physics,
and materials science.

# Acknowledgements

We are thankful to Yi-Lun Du for stimulating discussions and a detailed reading of the manuscript. We are grateful to Tian-Yao Hao, Ya Xu, and Qing-Yu You for their encouragement and providing computational resources. We acknowledge Fang-Zhou Nan and Yuan Wang for technical support. This research was supported in part by the National Natural Science Foundation of China through grants 41806083 and 91858212, and the AI grant at FIAS of SAMSON AG.

# A  Methods

## A.1  Generation of spin configurations

All random numbers were generated using a 32-bit Mersenne Twister pseudorandom number generator [25]. Monte Carlo simulations employing the Wolff algorithm [26] were used to generate spin configurations with periodic boundary conditions applied. Each lattice was first equilibrated for $2 \times 10^5$ cluster updates, after which spin configurations were sampled every $k = \lceil N/N_c \rceil$ cluster updates, where $N$ is the size of the lattice and $N_c$ is the average cluster size estimated from a separate simulation, and the value of $k$ is kept odd by adding one in cases where $\lceil N/N_c \rceil$ is even.

The sampling temperatures $T$ used in Monte Carlo simulations correspond to values that are regularly spaced on the finite-size scaled $x$ axis $(T - T_c)L^{1/\nu}$. Sampling temperatures were calculated in this manner in order to avoid biases in sampling using the same values of $T$ for all lattice sizes, where the number of configurations closer to the ordered and disordered extremes would increase, and samples available in the critical region would decrease, as $L$ increases. In particular, training data were generated in the interval $-16 \leq (T - T_c)L^{1/\nu} \leq 16$ with a resolution of 0.5, excluding $T_c$, to give a total of 64 sampling temperatures. Test data were generated in a similar manner using known values of $T_c$ and $\nu$; for systems that exhibit a first-order transition, the value of $\nu$ was chosen arbitrarily to produce a given amount of data points in the critical region.

## A.2  Neural network training

Ising model spin configurations with $N$ spins $\sigma$, where $\sigma \in \{-1, 1\}$, were converted to 1D arrays. Adjacent spins in a 1D array are also adjacent spins in the original lattice of higher dimension when no periodic boundary conditions are applied, and the next spin is always chosen with priority given to the next available spin along the $x$-axis, followed by the $y$-axis, and finally the $z$-axis. Spin configurations sampled at $T < T_c$, or the two representative ordered-state spin configurations of the simplified scheme, were labeled as $[1, 0]$. Similarly, spin configurations sampled at $T > T_c$, or the representative disordered-state spin configuration of the simplified scheme, were labeled as $[0, 1]$.

TensorFlow 1.12 [27] was used to construct and train all neural networks described. Arrays prepared as described above, 1200 samples at each temperature with $5 : 1$ training-validation split, were fed directly to ANNs that consist of an input layer of size $N$, a single hidden layer of 16 neurons activated by the SELU activation function, and an output layer of 2 sigmoid neurons. Weights were initialized with zero mean and a standard deviation of $N^{-\frac{1}{2}}$.

The training was performed with a batch size of 1% of the total number of samples, using an Adam optimizer with a learning rate of $1 \times 10^{-3}$ for the first 20 epochs, followed by a Nesterov Momentum optimizer with a momentum of 0.9 and the same learning rate for another 180 epochs. With the exception of the ANNs trained with the simplified state space that were

trained until losses were minimized, $L_2$ regularization was applied to minimize overfitting. The value of the regularization parameter $\lambda$ was determined dynamically at the beginning of training such that it would facilitate the condition $0.99 < loss_{training}/loss_{validation} < 1$ when training concludes at the $200^{th}$ epoch. For every lattice size, an ensemble of 10 neural networks was trained using 10 separate sets of training data.

### A.3   Classification of spin configurations

The transformation described in Eq. (1) was first applied to spin configurations before they were used to test the ANNs. For every temperature, $1 \times 10^5$ spin configurations were used to calculate $\mathcal{W}$; this procedure was performed for every member in an ensemble using the same test set. Ensemble averages and the associated 99.7% confidence intervals were used to produce the figures presented in this manuscript. Confidence intervals were calculated using the bootstrap method by resampling $1 \times 10^5$ times, and the larger magnitude of the two values produced was used in presentation and error propagation.

### A.4   Estimation of critical parameters

For each spin model examined, the critical temperature can be estimated from where curves cross on a plot of the output $\mathcal{W}$ against temperature, which allows the critical exponent $\nu$ to be estimated manually. These estimated values were used as the initial guesses for automated finite-size scaling using `autoScale.py` [28], and a grid search over values around these initial guesses and all other input parameters were then performed to minimize the output of the objective function. Standard errors were then estimated from the optimized values of $T'$ and $\nu'$. Critical temperatures for first-order transitions were estimated from a plot of temperature, at a given value of $\mathcal{W}$ that is close to where the curves cross, versus $1/L$ by standard extrapolation to $L \to \infty$; where $\mathcal{W} = 1/q$ for the regular ANNs, and $\mathcal{W} = 1.6/q$ for the ANNs trained with the simplified state space.

# B Summary of estimated critical properties

## B.1 Critical properties obtained using ANNs trained with Monte Carlo-sampled spin configurations (MC data)

Table 1: Critical properties of the ferromagnetic $q$-state Potts model obtained using ANNs trained with MC data. The values of $T'$ and $v'$ and their associated standard errors are estimated by finite-size scaling of ANN outputs. Literature values of $T_c$ and $v$ are included for comparison. *The values of $v'$ and $v$ [15] for the 4-state Potts model are without higher-order corrections.

| d | q | Geometry | $T'$ | $v'$ | $T_c$ | $v$ |
|---|---|----------|------|------|-------|-----|
| 2 | 2 | square | 1.1346(3) | 0.98(4) | 1.1346 | 1 |
| 2 | 3 | square | 0.99461(13) | 0.85(3) | 0.99497 | 0.833 |
| 2 | 4 | square | 0.91018(6) | 0.75(2) * | 0.91024 | 0.722 * |
| 2 | 5 | square | 0.85142(15) | - | 0.85153 | - |
| 2 | 6 | square | 0.80749(10) | - | 0.80761 | - |
| 2 | 7 | square | 0.77294(3) | - | 0.77306 | - |
| 2 | 2 | triangular | 1.8208(5) | 0.97(5) | 1.8205 | 1 |
| 2 | 3 | triangular | 1.5844(3) | 0.85(3) | 1.5849 | 0.833 |
| 2 | 4 | triangular | 1.4426(1) | 0.74(4) * | 1.4427 | 0.722 * |
| 2 | 5 | triangular | 1.3442(2) | - | 1.3445 | - |
| 2 | 6 | triangular | 1.2709(1) | - | 1.2714 | - |
| 2 | 7 | triangular | 1.2133(1) | - | 1.2140 | - |
| 2 | 2 | honeycomb | 0.75915(23) | 0.99(1) | 0.75933 | 1 |
| 2 | 3 | honeycomb | 0.67349(9) | 0.84(3) | 0.67376 | 0.833 |
| 2 | 4 | honeycomb | 0.62126(4) | 0.74(2) * | 0.62133 | 0.722 * |
| 2 | 5 | honeycomb | 0.58464(10) | - | 0.58474 | - |
| 2 | 6 | honeycomb | 0.55728(7) | - | 0.55720 | - |
| 2 | 7 | honeycomb | 0.53540(4) | - | 0.53544 | - |
| 3 | 2 | cubic | 2.2126(14) | 1.02(2) | 2.2558 | 0.630 |
| 3 | 3 | cubic | 1.8164(99) | - | 1.8163 | - |
| 3 | 4 | cubic | 1.5924(15) | - | 1.5908 | - |

Table 2: Critical properties of the antiferromagnetic 2-state Potts model obtained using ANNs trained with MC data. The values of $T'$ and $v'$ and their associated standard errors are estimated by finite-size scaling of ANN outputs. Literature values of $T_c$ and $v$ are included for comparison.

| d | q | Geometry | $T'$ | $v'$ | $T_c$ | $v$ |
|---|---|----------|------|------|-------|-----|
| 2 | 2 | square | 1.1344(3) | 1.00(1) | 1.1346 | 1 |

## B.2 Critical properties obtained using ANNs trained with representative spin configurations of the exponentially reduced state space (simplified data)

Table 3: Critical properties of the ferromagnetic $q$-state Potts model obtained using ANNs trained with simplified data. The values of $T'$ and $\nu'$ and their associated standard errors are estimated by finite-size scaling of ANN outputs. Literature values of $T_c$ and $\nu$ are included for comparison. * The values of $\nu'$ and $\nu$ [15] for the 4-state Potts model are without higher-order corrections.

| d | q | Geometry | $T'$ | $\nu'$ | $T_c$ | $\nu$ |
|---|---|----------|------|--------|-------|-------|
| 2 | 2 | square | 1.1364(12) | 1.01(11) | 1.1346 | 1 |
| 2 | 3 | square | 0.99561(69) | 0.84(8) | 0.99497 | 0.833 |
| 2 | 4 | square | 0.91042(20) | 0.73(7) * | 0.91024 | 0.722 * |
| 2 | 5 | square | 0.85144(53) | - | 0.85153 | - |
| 2 | 6 | square | 0.80739(38) | - | 0.80761 | - |
| 2 | 7 | square | 0.77301(14) | - | 0.77306 | - |
| 2 | 2 | triangular | 1.8234(19) | 1.00(17) | 1.8205 | 1 |
| 2 | 3 | triangular | 1.5857(8) | 0.85(14) | 1.5849 | 0.833 |
| 2 | 4 | triangular | 1.4429(3) | 0.74(7) * | 1.4427 | 0.722 * |
| 2 | 5 | triangular | 1.3445(6) | - | 1.3445 | - |
| 2 | 6 | triangular | 1.2717(5) | - | 1.2714 | - |
| 2 | 7 | triangular | 1.2142(1) | - | 1.2140 | - |
| 2 | 2 | honeycomb | 0.76040(71) | 1.02(5) | 0.75933 | 1 |
| 2 | 3 | honeycomb | 0.67422(38) | 0.85(8) | 0.67376 | 0.833 |
| 2 | 4 | honeycomb | 0.62152(21) | 0.74(8) * | 0.62133 | 0.722 * |
| 2 | 5 | honeycomb | 0.58468(60) | - | 0.58474 | - |
| 2 | 6 | honeycomb | 0.55738(35) | - | 0.55720 | - |
| 2 | 7 | honeycomb | 0.53540(20) | - | 0.53544 | - |
| 3 | 2 | cubic | 2.2502(31) | 0.68(31) | 2.2558 | 0.630 |
| 3 | 3 | cubic | 1.8188(107) | - | 1.8163 | - |
| 3 | 4 | cubic | 1.5903(6) | - | 1.5908 | - |

# C   Scatter plots of ANN output vs. temperature

## C.1   Ferromagnetic Potts models and ANNs trained with Monte Carlo-sampled spin configurations (MC data)

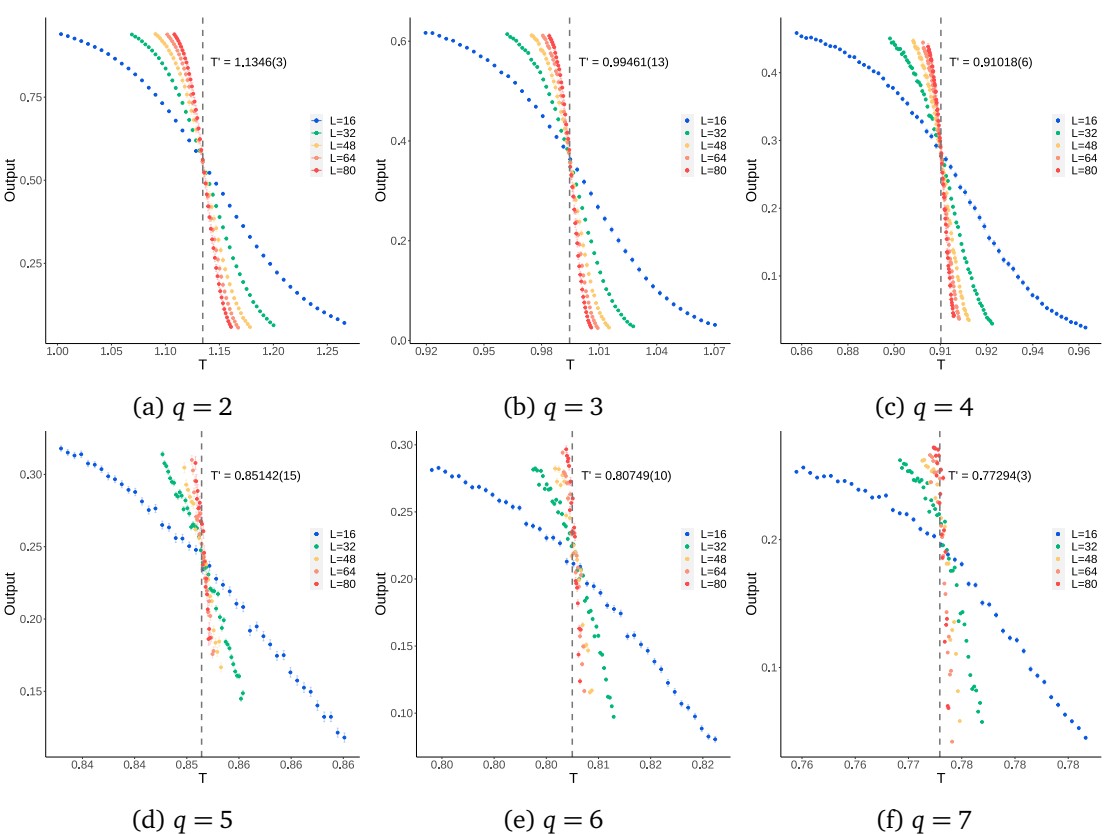

Figure 6: The relationship between temperature and the average output of an ensemble of 10 independently trained ANNs – training with MC data of the ferromagnetic square-lattice Ising model, and testing with MC data of the ferromagnetic $q$-state Potts model on a square lattice. The error bars shown are 99.7% confidence intervals of the ensemble average.

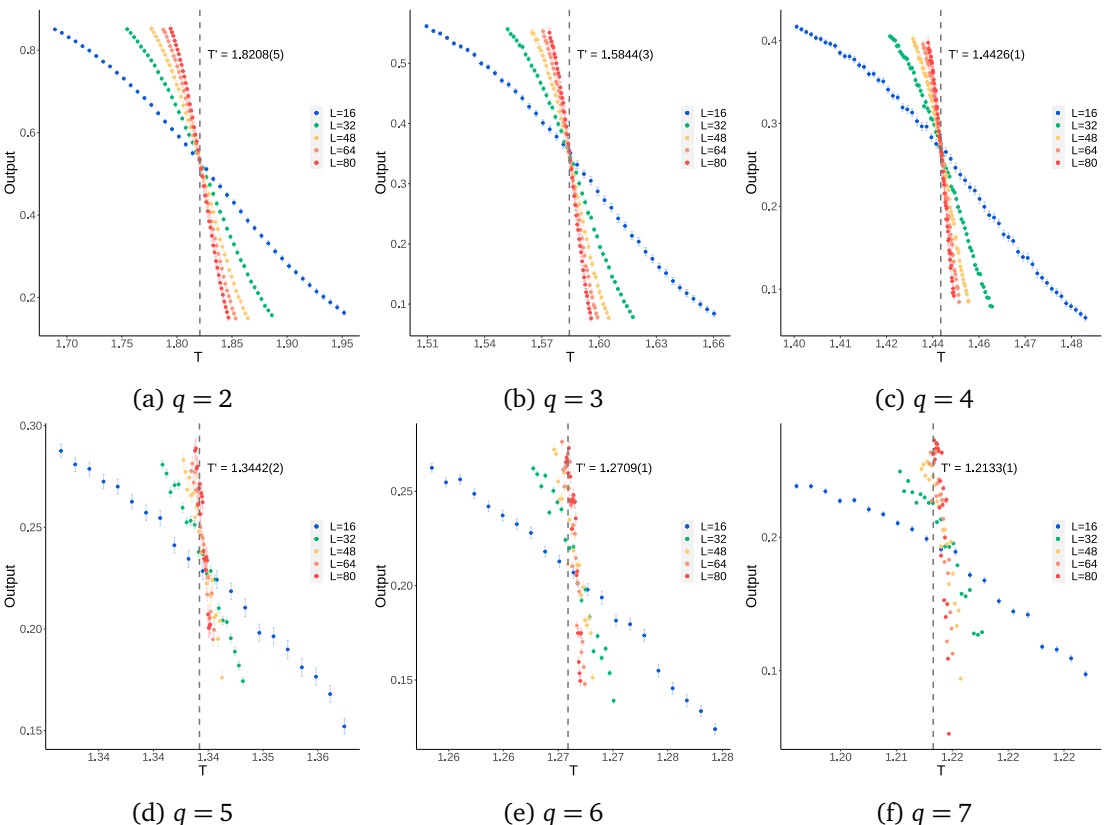

Figure 7: The relationship between temperature and the average output of an ensemble of 10 independently trained ANNs – training with MC data of the ferromagnetic square-lattice Ising model, and testing with MC data of the ferromagnetic $q$-state Potts model on a triangular lattice. The error bars shown are 99.7% confidence intervals of the ensemble average.

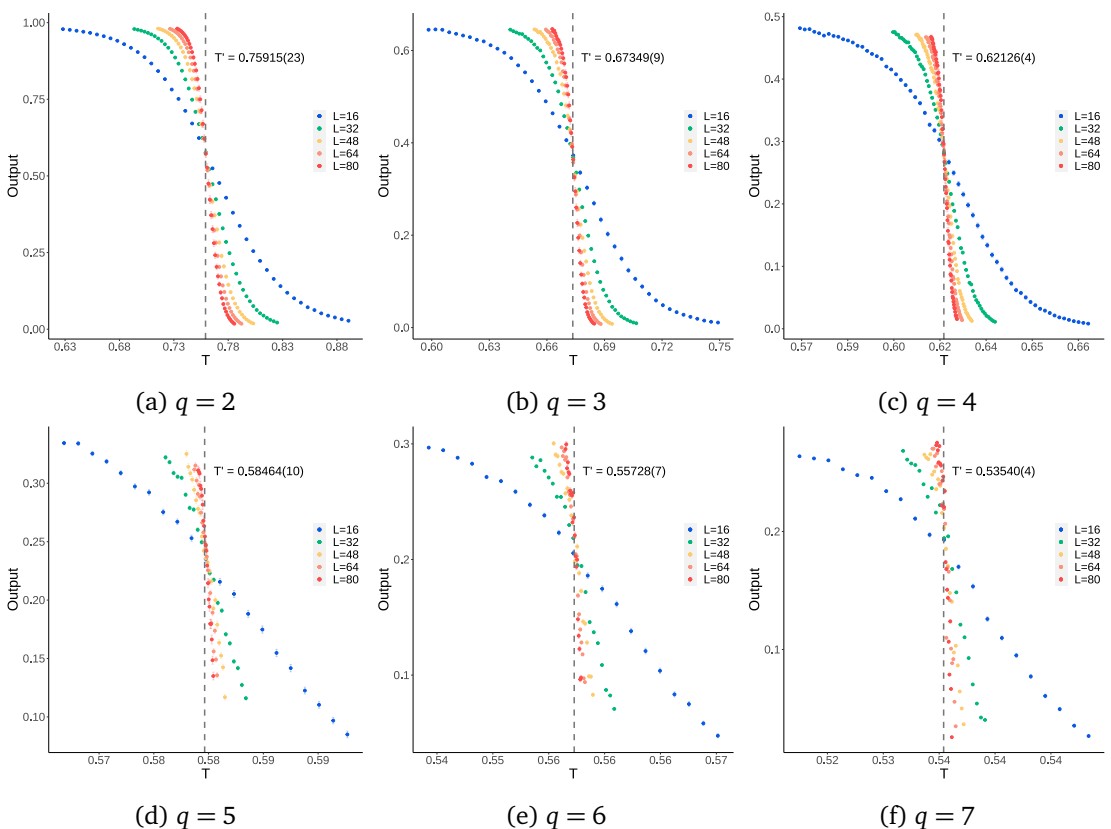

Figure 8: The relationship between temperature and the average output of an ensemble of 10 independently trained ANNs – training with MC data of the ferromagnetic square-lattice Ising model, and testing with MC data of the ferromagnetic $q$-state Potts model on a honeycomb lattice. The error bars shown are 99.7% confidence intervals of the ensemble average.

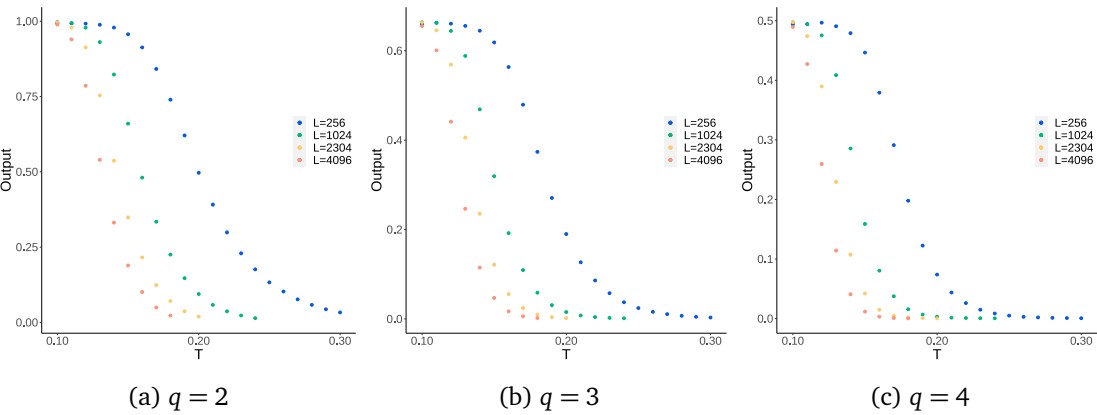

Figure 9: The relationship between temperature and the average output of an ensemble of 10 independently trained ANNs – training with MC data of the ferromagnetic square-lattice Ising model, and testing with MC data of the one-dimensional ferromagnetic $q$-state Potts model. The error bars shown are 99.7% confidence intervals of the ensemble average.

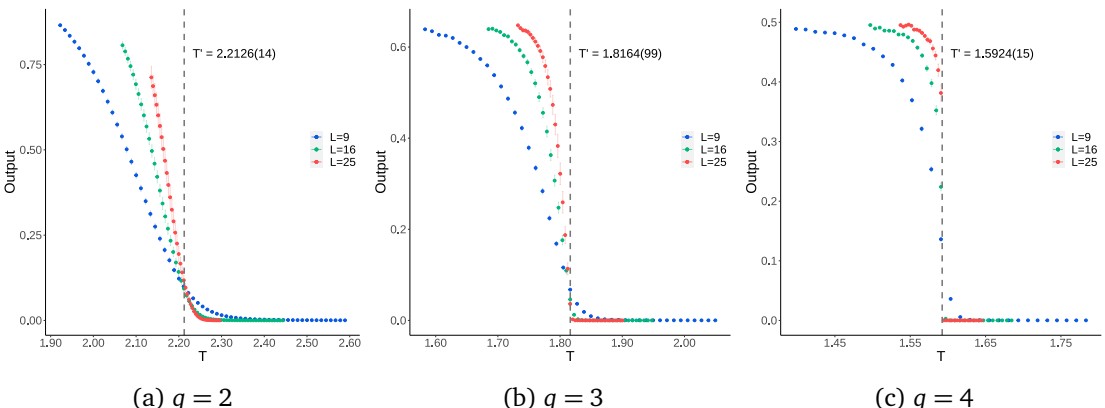

Figure 10: The relationship between temperature and the average output of an ensemble of 10 independently trained ANNs – training with MC data of the ferromagnetic square-lattice Ising model, and testing with MC data of the ferromagnetic $q$-state Potts model on a cubic lattice. The error bars shown are 99.7% confidence intervals of the ensemble average.

## C.2 Antiferromagnetic Potts models and ANNs trained with Monte Carlo-sampled spin configurations (MC data)

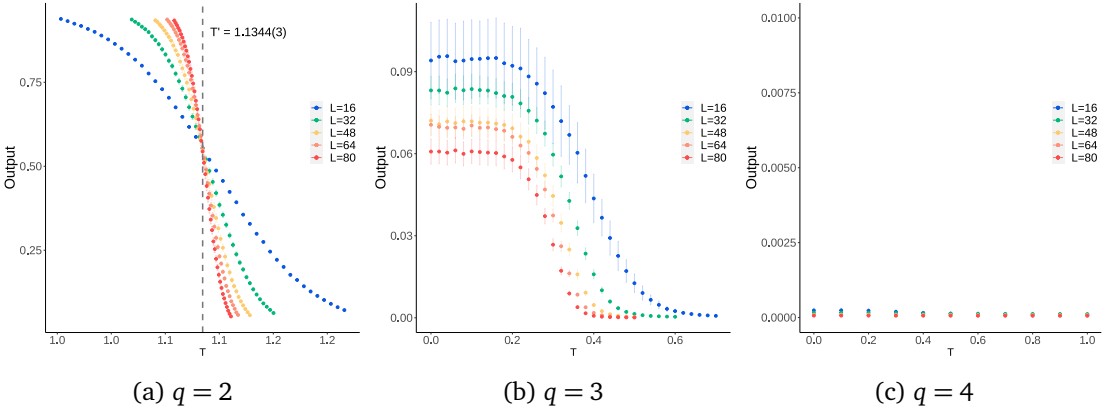

Figure 11: The relationship between temperature and the average output of an ensemble of 10 independently trained ANNs – training with MC data of the antiferromagnetic square-lattice Ising model, and testing with MC data of the antiferromagnetic $q$-state Potts model on a square lattice. The error bars shown are 99.7% confidence intervals of the ensemble average.

### C.3 Ferromagnetic Potts models and ANNs trained with representative spin configurations of the exponentially reduced state space (simplified data)

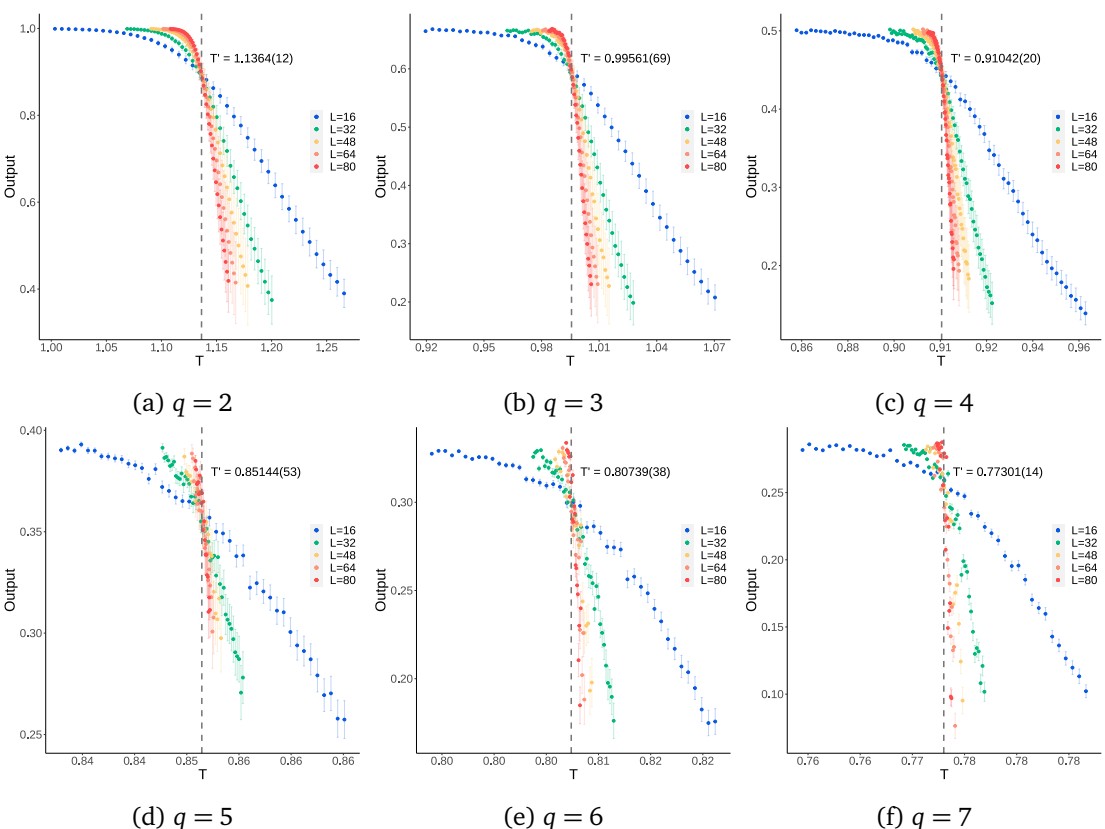

Figure 12: The relationship between temperature and the average output of an ensemble of 10 independently trained ANNs – training with simplified data, and testing with MC data of the ferromagnetic $q$-state Potts model on a square lattice. The error bars shown are 99.7% confidence intervals of the ensemble average.

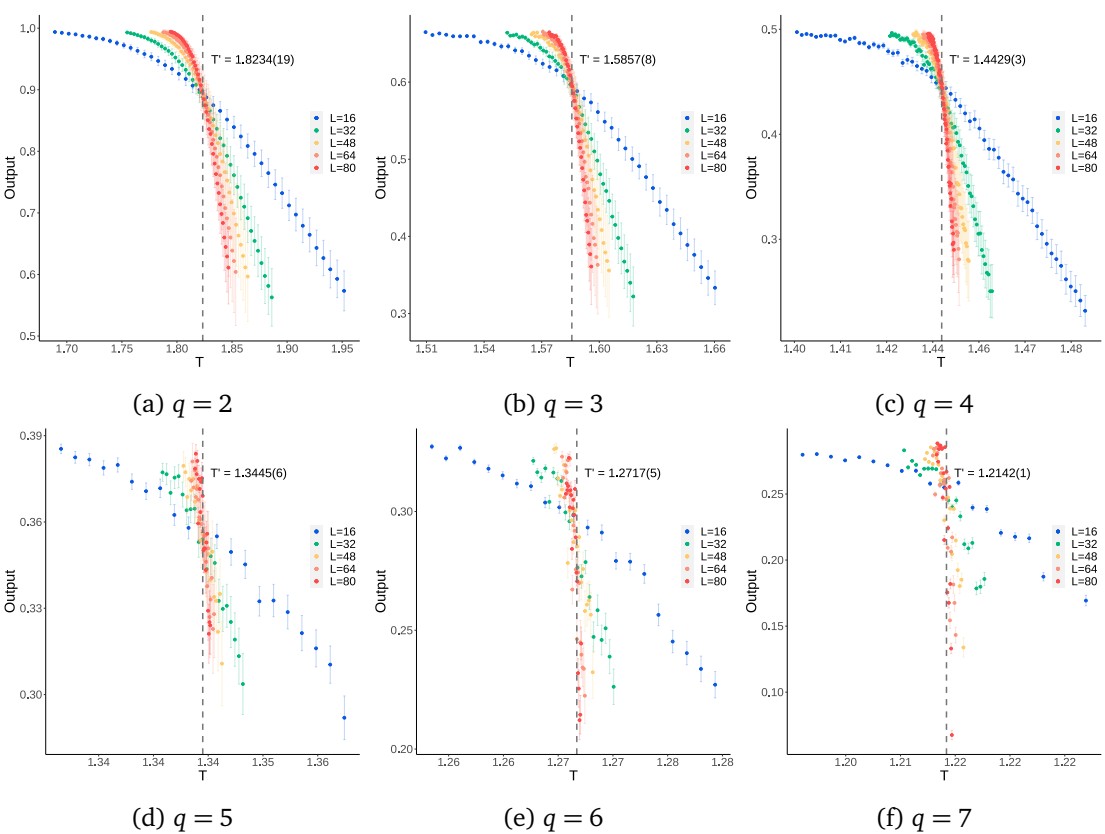

Figure 13: The relationship between temperature and the average output of an ensemble of 10 independently trained ANNs – training with simplified data, and testing with MC data of the ferromagnetic $q$-state Potts model on a triangular lattice. The error bars shown are 99.7% confidence intervals of the ensemble average.

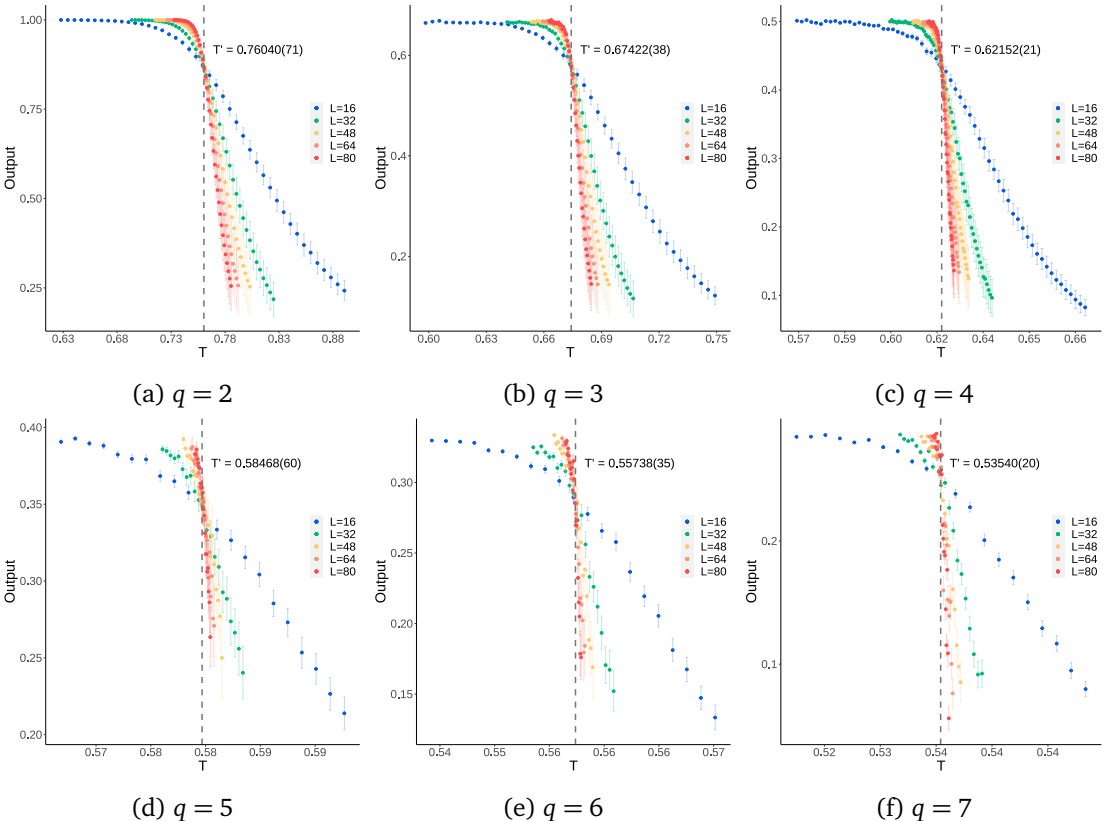

Figure 14: The relationship between temperature and the average output of an ensemble of 10 independently trained ANNs – training with simplified data, and testing with MC data of the ferromagnetic $q$-state Potts model on a honeycomb lattice. The error bars shown are 99.7% confidence intervals of the ensemble average.

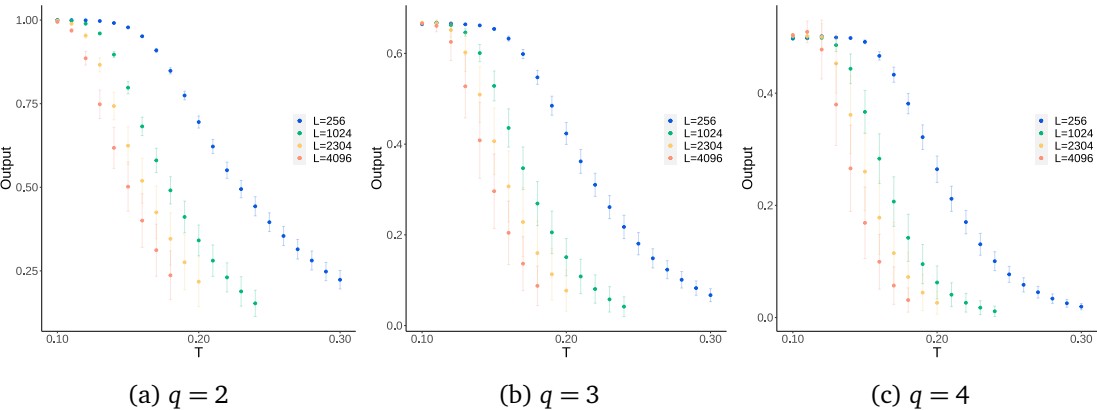

Figure 15: The relationship between temperature and the average output of an ensemble of 10 independently trained ANNs – training with simplified data, and testing with MC data of the one-dimensional ferromagnetic $q$-state Potts model. The error bars shown are 99.7% confidence intervals of the ensemble average.

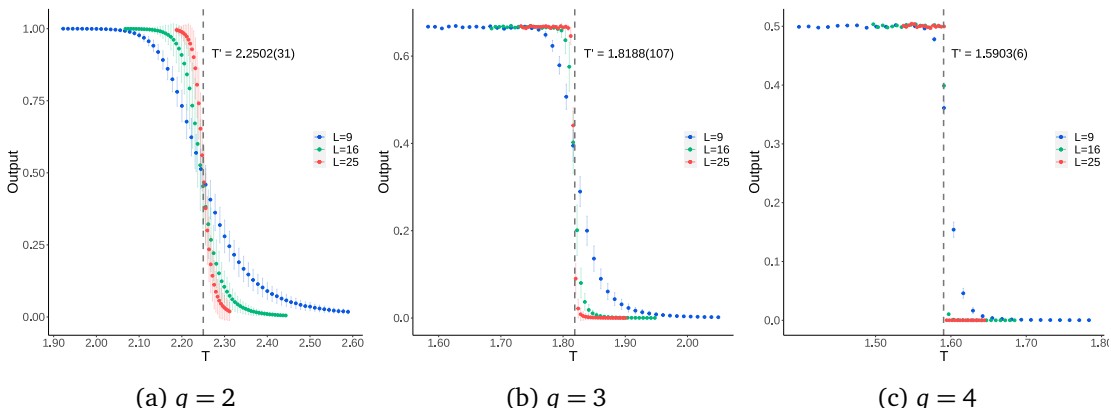

(a) $q = 2$      (b) $q = 3$      (c) $q = 4$

Figure 16: The relationship between temperature and the average output of an ensemble of 10 independently trained ANNs – training with simplified data, and testing with MC data of the ferromagnetic $q$-state Potts model on a cubic lattice. The error bars shown are 99.7% confidence intervals of the ensemble average.

# D    Scatter plots with finite-size scaling

## D.1    Ferromagnetic Potts models and ANNs trained with Monte Carlo-sampled spin configurations (MC data)

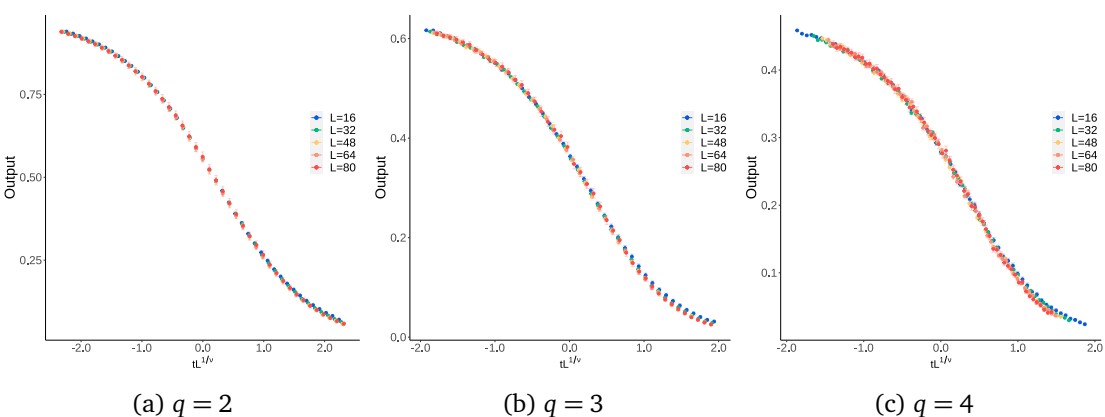

(a) $q = 2$      (b) $q = 3$      (c) $q = 4$

Figure 17: The finite-size scaled plots of the average output of an ensemble of 10 independently trained ANNs – training with MC data of the ferromagnetic square-lattice Ising model, and testing with MC data of the ferromagnetic $q$-state Potts model on a square lattice. The error bars shown are 99.7% confidence intervals of the ensemble average.

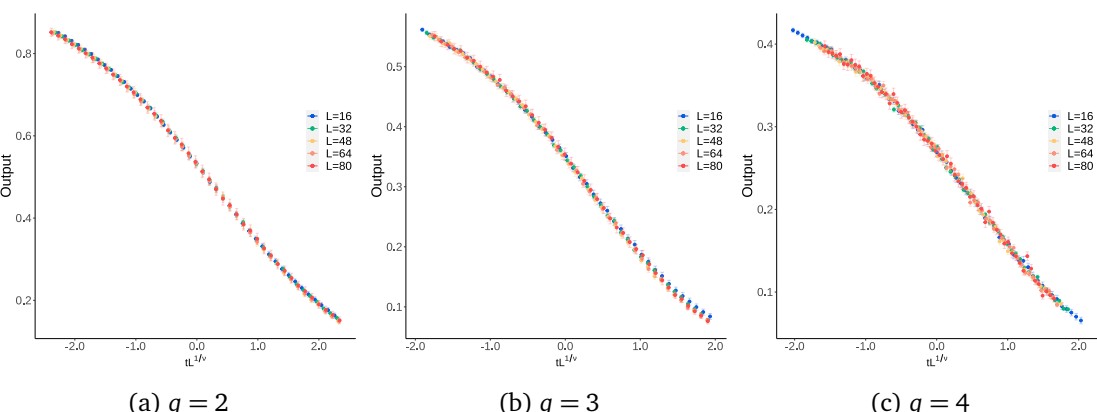

(a) $q = 2$       (b) $q = 3$       (c) $q = 4$

Figure 18: The finite-size scaled plots of the average output of an ensemble of 10 independently trained ANNs – training with MC data of the ferromagnetic square-lattice Ising model, and testing with MC data of the ferromagnetic $q$-state Potts model on a triangular lattice. The error bars shown are 99.7% confidence intervals of the ensemble average.

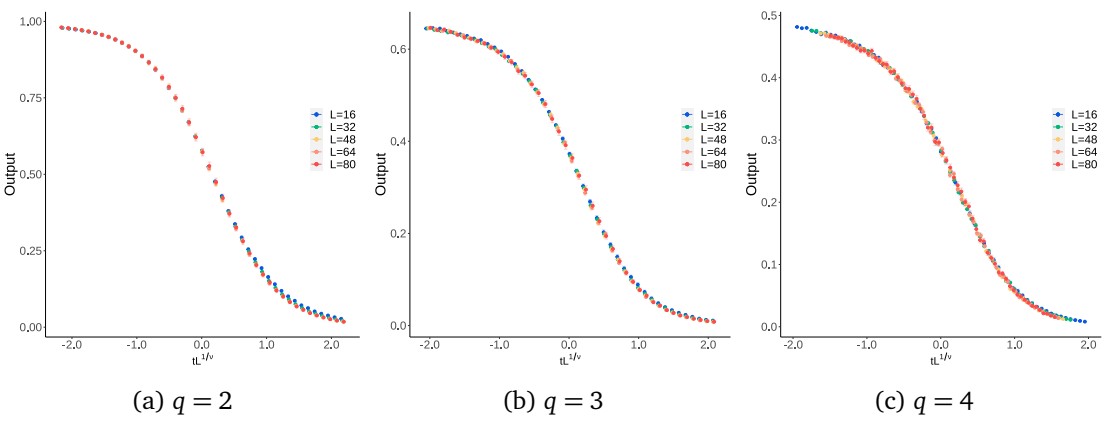

(a) $q = 2$       (b) $q = 3$       (c) $q = 4$

Figure 19: The finite-size scaled plots of the average output of an ensemble of 10 independently trained ANNs – training with MC data of the ferromagnetic square-lattice Ising model, and testing with MC data of the ferromagnetic $q$-state Potts model on a honeycomb lattice. The error bars shown are 99.7% confidence intervals of the ensemble average.

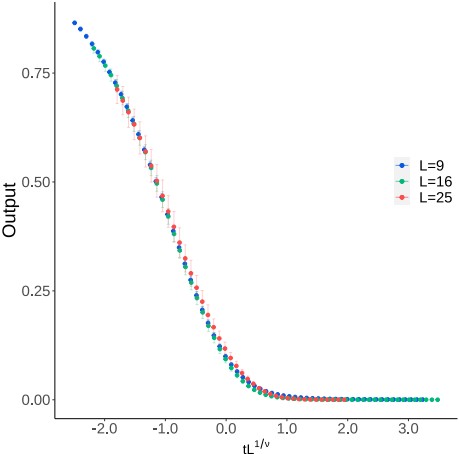

Figure 20: $q = 2$: The finite-size scaled plot of the average output of an ensemble of 10 independently trained ANNs – training with MC data of the ferromagnetic square-lattice Ising model, and testing with MC data of the ferromagnetic 2-state Potts model on a cubic lattice. The error bars shown are 99.7% confidence intervals of the ensemble average.

## D.2 Antiferromagnetic Potts models and ANNs trained with Monte Carlo-sampled spin configurations (MC data)

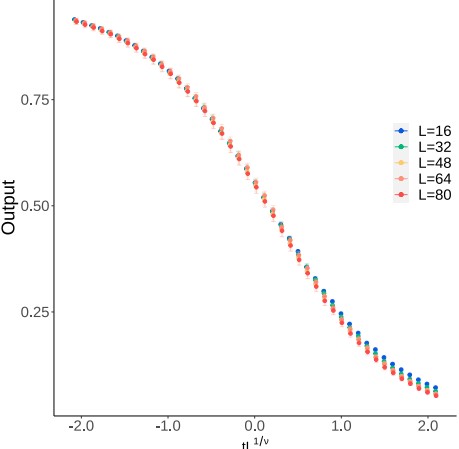

Figure 21: $q = 2$: The finite-size scaled plot of the average output of an ensemble of 10 independently trained ANNs – training with MC data of the antiferromagnetic square-lattice Ising model, and testing with MC data of the antiferromagnetic 2-state Potts model on a square lattice. The error bars shown are 99.7% confidence intervals of the ensemble average.

## D.3 Ferromagnetic Potts models and ANNs trained with representative spin configurations of the exponentially reduced state space (simplified data)

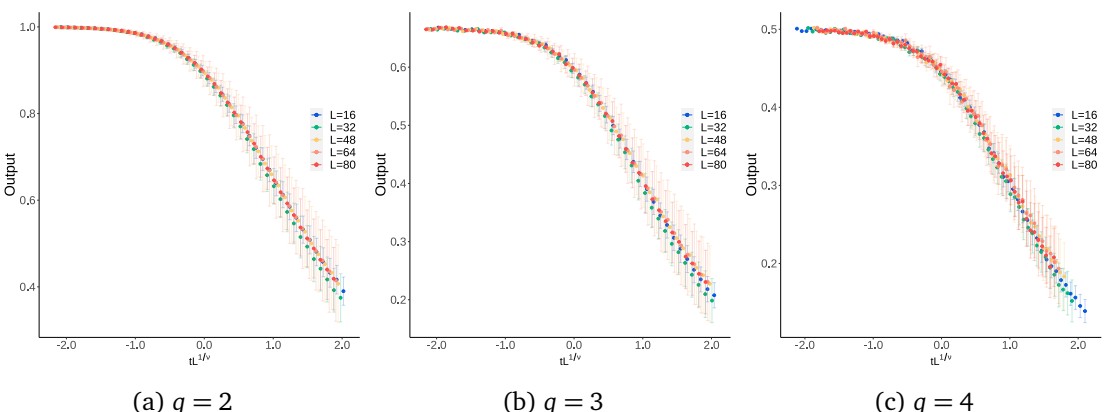

(a) $q = 2$      (b) $q = 3$      (c) $q = 4$

Figure 22: The finite-size scaled plots of the average output of an ensemble of 10 independently trained ANNs – training with simplified data, and testing with MC data of the ferromagnetic $q$-state Potts model on a square lattice. The error bars shown are 99.7% confidence intervals of the ensemble average.

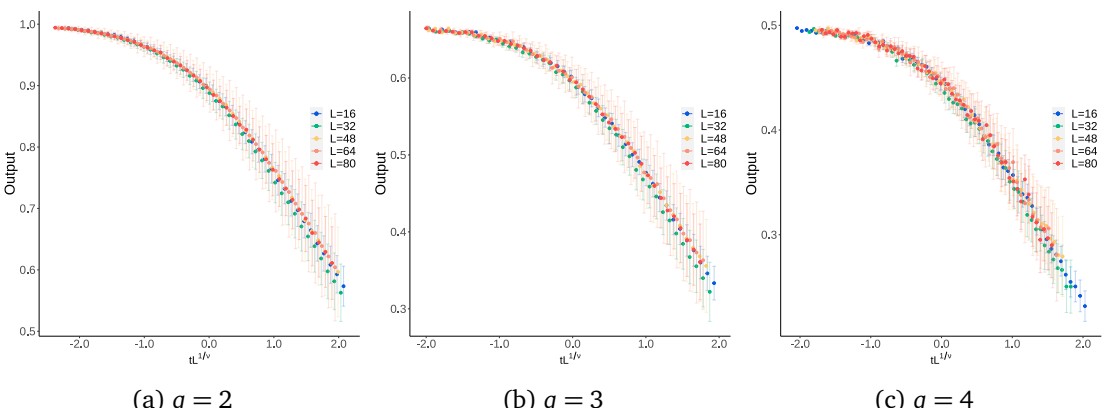

(a) $q = 2$      (b) $q = 3$      (c) $q = 4$

Figure 23: The finite-size scaled plots of the average output of an ensemble of 10 independently trained ANNs – training with simplified data, and testing with MC data of the ferromagnetic $q$-state Potts model on a triangular lattice. The error bars shown are 99.7% confidence intervals of the ensemble average.

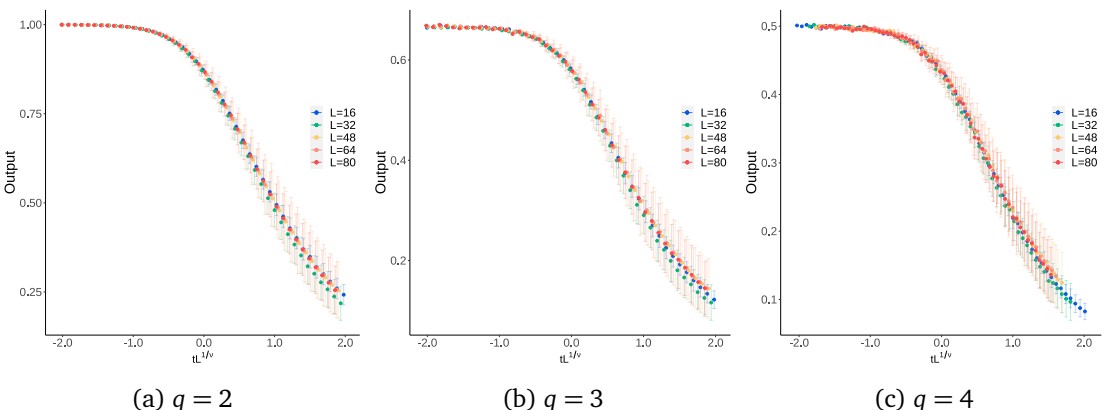

Figure 24: The finite-size scaled plots of the average output of an ensemble of 10 independently trained ANNs – training with simplified data, and testing with MC data of the ferromagnetic $q$-state Potts model on a honeycomb lattice. The error bars shown are 99.7% confidence intervals of the ensemble average.

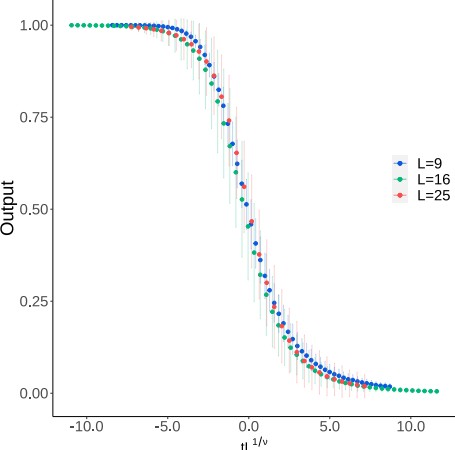

Figure 25: $q = 2$: The finite-size scaled plot of the average output of an ensemble of 10 independently trained ANNs – training with simplified data, and testing with MC data of the ferromagnetic 2-state Potts model on a cubic lattice. The error bars shown are 99.7% confidence intervals of the ensemble average.

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
