# Peer review of "On the generalizability of artificial neural networks in spin models"

_SciPost Physics, doi:SciPost Phys. Core 5, 032 (2022)_

## Round 1 · Referee Report · Anonymous (Referee 2) · 2021-12-14

Strengths

The work is well structured and written, so it could contribute to the knowledge of the power of generalization of ANNs in condensed matter.

Weaknesses

The issue of generalizability in classical spin models has been studied in several works. Lack of proper references to some of these works.

Report

This work explores the possibilities of artificial neural networks (ANN) to classify phases and their transitions beyond the context in which they have been trained.

The work begins with the training of ANNs to detect the order-disorder transition in the ferromagnetic Ising model in the square lattice. This ANN is then used to determine phase transitions in the q-state Potts model, for different q values, lattice geometries, and dimensions. Then the authors apply the same scheme for the antiferromagnetic case. Finally, a simplified model is explored with a minimum number of training configurations, which captures the essential aspects previously studied. Although the issue of generalizability in classical spin models has been studied in several works, the authors propose a novel mapping, which allows them to determine and compare transition temperatures and critical exponents (where appropriate).

The work is well structured and written, so it could contribute to the knowledge of the power of generalization of ANNs in condensed matter. However, before recommending this work for publication, I would like the authors to reply and clarify the following aspects:

1) Due to the proposed mapping (Eq. 1 of the paper), the maximum value that the ANN output takes is $2/3$ for $q = 3$, $1/2$ for $q = 4$, and so on. Although the reason is clear, it is impractical to read the output with a probability whose maximum value should be 1 in the ordered phase. Since this is just a normalization factor, I suggest rescaling the output such that its maximum value is 1 in all cases.

2) In cases where there are second-order transitions, crossovers of the ANN outputs occur for different lattice sizes. In the work it is assumed that these crosses indicate the transition temperature and the scale invariance in criticality (which is not obvious, nor direct, but numerically observable in the figures). Regarding this scaling I think that in the scaled curves (for example Figs. 1 d) e) f)) the x-axis would not be $t$ but $t L ^ {1 / \nu}$?

3) When the transition is first order (as in the cases of the Potts model with $q >4$) there is no crossing in the transition. It could be useful to clarify how the transition temperature is determined in such cases (eg the determinations in figs. 6 d, e, f)?

4) Regarding the simplified model trained with only three configurations, I suppose that this reduction is also reflected in the structure of the weights and biases of the ANN. It would be interesting to compare this simplification with the toy model proposed in the work of Carrasquilla and Melko (ref 2 in the paper). There, the reduction is carried out over the structure of the ANN itself, using only three hidden neurons with an analytical expression of the weights and the biases (with a single adjustable parameter). I suspect that the simplification that the authors propose can be made even more transparent by inspecting the structure of the ANN as is done in the work I mention.

5) The simplified model presented in the paper works for the ferromagnetic case. I suppose it is possible to generate a similar model for the antiferromagnetic case. However, I would like to know the opinion of the authors about the implementation of these "minimal" training models, for more complex cases, for example incorporating other interactions beyond first neighbors, or external fields.

  • validity: good
  • significance: good
  • originality: ok
  • clarity: good
  • formatting: good
  • grammar: good

Author:  Nan Su  on 2022-03-20  [id 2306]

(in reply to Report 2 on 2021-12-14)
Category:
answer to question

We thank the referee for the report. Please find our reply to each questions in the follow.

1) Due to the proposed mapping (Eq. 1 of the paper), the maximum value that the ANN output takes is 2/3 for q=3, 1/2 for q=4, and so on. Although the reason is clear, it is impractical to read the output with a probability whose maximum value should be 1 in the ordered phase. Since this is just a normalization factor, I suggest rescaling the output such that its maximum value is 1 in all cases.

We thank the referee for raising this important issue. We had originally considered normalizing the output; however, we opted to show the raw data in the submitted manuscript for clarity and to highlight the importance of this scaling — we think that the “exact” scaling of 2/q would not exist if our assumptions about the generalization we see were incorrect. We have revised the Manuscript according to the referee’s suggestion, and also included the original in the Appendix for comparison.

2) In cases where there are second-order transitions, crossovers of the ANN outputs occur for different lattice sizes. In the work it is assumed that these crosses indicate the transition temperature and the scale invariance in criticality (which is not obvious, nor direct, but numerically observable in the figures). Regarding this scaling I think that in the scaled curves (for example Figs. 1 d) e) f)) the x-axis would not be t but tL^{1/ν}?

We thank the referee for pointing out this typo, and the axis label for the plots in question should indeed be tL^{1/ν}. We have corrected this typographic mistake in the revised Manuscript.

3) When the transition is first order (as in the cases of the Potts model with q>4) there is no crossing in the transition. It could be useful to clarify how the transition temperature is determined in such cases (eg the determinations in figs. 6 d, e, f)?

The relevant discussion on how we calculated the transition temperature for the first order transition was included in the submitted Manuscript in Appendix A.4 Estimation of critical parameters, which we did not mention explicitly in the main text. In order to improve the discoverability of this information, we have added the following sentence

“see Appendix A.4 for a description of how the corresponding critical parameters are determined”

in the relevant paragraph on page 4.

4) Regarding the simplified model trained with only three configurations, I suppose that this reduction is also reflected in the structure of the weights and biases of the ANN. It would be interesting to compare this simplification with the toy model proposed in the work of Carrasquilla and Melko (ref 2 in the paper). There, the reduction is carried out over the structure of the ANN itself, using only three hidden neurons with an analytical expression of the weights and the biases (with a single adjustable parameter). I suspect that the simplification that the authors propose can be made even more transparent by inspecting the structure of the ANN as is done in the work I mention.

Our simplified model offers a different ANN modeling to the one proposed in the work of Carrasquilla and Melko whose construction focus on modeling the ANN structure. In contrast, ours focus on modeling the training data, which is simply a straightforward utilization of the known exact ground state [-1,-1,-1,…] and [1,1,1,…], as well as the intuitive observation of [0,0,0,…] representing the disordered phase. Although in this way the training data may be kept minimal, before carrying out the numerical experiments, there was no guarantee whatsoever that such a counterintuitive approach should work, since such a few-data training strategy was totally against the conventional wisdom of large amounts of training data for the same task.

The structure of the ANN may reveal deeper insights about why our unconventional training strategy works as the referee suspected, which we plan to systematically tackle in a next work. Since the numerical experiments with the simplified model were systematic and the results were very surprising, especially considering its potential impact in accelerating the development times and saving computing resources of related machine learning tasks, we felt it would be timely to first report the results in the current manuscript. For this reason, we have added the following sentence to the Conclusions on page 9:

“An analysis of the ANN structures may reveal deeper insights about why such a minimal training strategy works, and this reduction may introduce simplification in the same or similar tasks in quantum machine learning. We plan to pursue these in a future work.”

5) The simplified model presented in the paper works for the ferromagnetic case. I suppose it is possible to generate a similar model for the antiferromagnetic case. However, I would like to know the opinion of the authors about the implementation of these "minimal" training models, for more complex cases, for example incorporating other interactions beyond first neighbors, or external fields.

We would like to thank the referee for taking a keen interest in our work! Heuristics guided the development of the simplified model for the ferromagnetic case, where 0 in the training data effectively means “randomized”. For this reason, we believe in the methodology independent of the details of the system, such as dimensionality, interaction types, and external fields, and we are positive about the same construction for more complex models such as those involving, but not limited to, antiferromagnetism.

---

## Round 1 · Referee Report · Carlos Lamas (Referee 1) · 2021-12-14

Strengths

1- The subject matter is interesting and timely. 2- The authors attempt some approaches that could be novel.

Weaknesses

1- The authors make no effort to understand why classification works in the case studies even though they have the tools to do so.

2-Comparison with previous literature results in similar systems is poor. This does not make it possible to see clearly what is new in each of the results presented in the manuscript.

Report

In the manuscript entitled “On the generalizability of artificial neural networks in spin models” the authors present a study on the generalizability of simple artificial neural networks (ANN).
The authors have trained the ANN with the two-dimensional ferromagnetic Ising model and then successfully apply to the ferromagnetic q-state Potts model in different dimensions.

The paper is interesting and well written, and I think this work may be interesting, but before recommending this work, the manuscript needs some corrections and clarifications. In particular, the authors should discuss similarities and differences with previous results using the same techniques to identify phases in frustrated Ising-type models in order to make clear what is the novel contribution of the work.

Comments:

The study of the generalization capability of ANNs was already studied in previous works; in particular the generalization in antiferromagnetic systems, in the presence of frustration and change of the lattice geometry was studied in [1][2]. Authors should emphasize what is the substantial difference between the study presented in the manuscript and that studied in those references.

The authors performed supervised learning on fully-connected feed-forward ANNs consisting in a single hidden layer of 16 neurons and 2 output neurons for carrying out the binary classification of spin configurations. The 2 output neurons are standard in the classification of spin configurations and the 2 outputs are usually interpreted as the probability that the input configuration belongs to the ordered or disordered phase. In this paper the authors refer throughout the paper to an output W, which I interpret as the probability of belonging to the ordered phase. This output W is plotted in Figure 1-a for different sizes, where a crossover of all curves at a given temperature value can be observed. On the one hand, the scaling used in the manuscript had already been used by Carrasquilla et al [1]. On the other hand, the authors argue that this is the transition temperature because of scale invariance. However, the ANN output is not a physical observable, so why do the authors expect this quantity to exhibit scale invariance? And more than this, why do they expect the exponents to be the same as those in the Potts model?

In light of previous results on the performance of ANNs in the study of the antiferromagnetic Ising model, the results presented in section 2.1.3 are not surprising. In this sense I suggest to change the title “a nontrivial exploration”. In fact, In this section It would be much more interesting to study the ANN performance in the presence of frustration.

In section 2.2, the authors trained ANNs with only three artificial spin configurations, {1, 1, . . . , 1}, {−1, −1, . . . , −1}, representing the ordered phase; and the spin configuration {0, 0, . . . , 0} representing the disordered phase. In this way, the dimension of the data set is exponentially reduced.
This reduction is potentially interesting, but I see no effort on the part of the authors to understand why the neural network can correctly classify phases with such a reduced training set. By training with so little data, the weight structure of the ANN must be highly redundant. I think the authors should study the ANN weights to understand this point.

[1] Nature Phys 13, 431–434 (2017)
[2] Computational Materials Science 198, 110702 (2021)
  • validity: good
  • significance: ok
  • originality: ok
  • clarity: good
  • formatting: good
  • grammar: excellent

Author:  Nan Su  on 2022-03-20  [id 2307]

(in reply to Report 1 by Carlos Lamas on 2021-12-14)
Category:
answer to question

We thank the referee for the report. Please find our reply to each questions in the follow.

1) The study of the generalization capability of ANNs was already studied in previous works; in particular the generalization in antiferromagnetic systems, in the presence of frustration and change of the lattice geometry was studied in [1][2]. Authors should emphasize what is the substantial difference between the study presented in the manuscript and that studied in those references.

As we have stated in the relevant parts in the Manuscript the generalizability we have established in the current study is a systematic method for the application of ANNs trained with a simple model (i.e. with low symmetry) at a low dimension to the identification of critical phenomena of potentially an infinite series of complex models (i.e. with higher symmetries) irrespective of dimensionality.

This is a new type of generalizability than the generalizability to different geometries within the same model or symmetry as reported in [1]. This is also in sharp contrast to the generalizability reported in [2] as the two studies focus on totally different aspects: [2] explores the ANN generalizability for frustrated spin models, and ours is a systematic study of the generalizability of ANNs in non-frustrated systems by applying ANNs trained with one simple model to an infinite series of complex models obeying certain symmetries. To the best of our knowledge, our study is the first of its kind in addressing this new type of ANN generalizability in spin models. In order to help the audience to better comprehend this subtlety as the referee suggested, we have revised the discussions in the Introduction to the following (Pages 2-3):

“This novel generalizability is different than the ANN generalizability to different lattice geometries within the same model or symmetry reported in Refs. [2,6], as it significantly enlarges the applicability of the trained ANNs to unseen, nontrivial problems, especially in the study of phases and critical phenomena of matter and materials described by the Potts model [13]. It is also in contrast to the one reported in Ref. [14] as the two studies focus on different aspects: while Ref. [14] explores the ANN generalizability for frustrated spin models, ours tackles at the level of systematics the ANN generalizability for non-frustrated systems.”

2) The authors performed supervised learning on fully-connected feed-forward ANNs consisting in a single hidden layer of 16 neurons and 2 output neurons for carrying out the binary classification of spin configurations. The 2 output neurons are standard in the classification of spin configurations and the 2 outputs are usually interpreted as the probability that the input configuration belongs to the ordered or disordered phase. In this paper the authors refer throughout the paper to an output W, which I interpret as the probability of belonging to the ordered phase. This output W is plotted in Figure 1-a for different sizes, where a crossover of all curves at a given temperature value can be observed. On the one hand, the scaling used in the manuscript had already been used by Carrasquilla et al [1]. On the other hand, the authors argue that this is the transition temperature because of scale invariance. However, the ANN output is not a physical observable, so why do the authors expect this quantity to exhibit scale invariance? And more than this, why do they expect the exponents to be the same as those in the Potts model?

Our methodology of extracting the critical quantities is listed in the Appendix A.4 Estimation of critical parameters. We follow the same, well-established finite size scaling ansatz as reported in [1], which has has been shown to be related to magnetization, to extract the critical temperature and the critical exponent nu. It is worth noting that this methodology has also been widely adopted by work that builds upon the findings of [1] in the literature.

3) In light of previous results on the performance of ANNs in the study of the antiferromagnetic Ising model, the results presented in section 2.1.3 are not surprising. In this sense I suggest to change the title “a nontrivial exploration”. In fact, In this section It would be much more interesting to study the ANN performance in the presence of frustration.

As we have explained above that our study focuses on different generalizability than [2]. The antiferromagnetic exploration we carried out focuses on the systematic application of ANNs trained with antiferromagnetic Ising model with two-fold degenerate ground states to the identification of critical phenomena of antiferromagnetic Potts models with infinitely degenerate ground states. We are not aware of any previous work utilizing ANNs trained with finite degenerate systems in the classification task to infinitely degenerate systems, which justifies the use of “nontrivial” in the title. Therefore we believe that the use of “nontrivial” is appropriate in the context condensed matter physics and prefer to keep the original title unchanged.

4) In section 2.2, the authors trained ANNs with only three artificial spin configurations, {1, 1, . . . , 1}, {−1, −1, . . . , −1}, representing the ordered phase; and the spin configuration {0, 0, . . . , 0} representing the disordered phase. In this way, the dimension of the data set is exponentially reduced. This reduction is potentially interesting, but I see no effort on the part of the authors to understand why the neural network can correctly classify phases with such a reduced training set. By training with so little data, the weight structure of the ANN must be highly redundant. I think the authors should study the ANN weights to understand this point.

As the referee have pointed out the timeliness of our study, we felt exactly the same when preparing the Manuscript. Although our simplified model may offer a strategy with minimal training data, before carrying out the numerical experiments, there was no guarantee whatsoever that such a counterintuitive approach should work, since such a few-shot training strategy was totally against the conventional wisdom of large amounts of training data for the same task. Since the numerical experiments with the simplified model are systematic and the results were very surprising, especially considering its potential impact in accelerating the development times and saving computing resources of related machine learning tasks, we felt it would be timely to first report the results in the current Manuscript.

We agree with the referee that the reduction is potentially interesting and that studying the structure of the ANN may reveal deeper insights about why our unconventional training strategy works. The systematic and statistical analysis of the ANN structure is time consuming which worths a separate publication and we plan to tackle it in a future work. We have added the following sentence to the Conclusions on page 9:

“An analysis of the ANN structures may reveal deeper insights about why such a minimal training strategy works, and this reduction may introduce simplification in the same or similar tasks in quantum machine learning. We plan to pursue these in a future work.”

---

## Round 2 · Referee Report · Anonymous (Referee 3) · 2022-3-23

Report

In the new version of the manuscript, the authors have added some sentences, but have not satisfactorily responded to comments on the interpretation of the results. The work is correct, but it represents a systematic and unprofound study of the functioning of a neural network. Aspects with physical content, such as the appearance of a scale invariance for the output of the network, are not studied in depth. For these reasons, I believe that the manuscript does not meet the strict acceptance criteria for SciPost Physics and would be more suitable for SciPost Physics Core.

---

## Round 2 · Referee Report · Anonymous (Referee 2) · 2022-3-23

Report

Dear authors
Thank you for the answers to the issues and questions raised.
I consider that the paper in its current state is suitable for publication.
However, given the general characteristics of the work in its present
form, and taking into account the acceptance criteria of both SciPost
Physics and SciPost Physics Core, I consider the latter to be more suitable for publishing the work.

---

## Round 2 · Author Response

Dear Editor,

We would like to thank the referees for their reports. We have addressed them in the replies, and we hope our revised version is now ready for publication.

Best regards,
Hon Man Yau, Nan Su

---

## Round 2 · List of Changes

1) Ref.[14] added

2) All plots in the Manuscript updated to q/2-normalized ones

3) The following sentences added in pages 2-3:

“This novel generalizability is different than the ANN generalizability to different lattice geometries within the same model or symmetry reported in Refs. [2,6], as it significantly enlarges the applicability of the trained ANNs to unseen, nontrivial problems, especially in the study of phases and critical phenomena of matter and materials described by the Potts model [13]. It is also in contrast to the one reported in Ref. [14] as the two studies focus on different aspects: while Ref. [14] explores the ANN generalizability for frustrated spin models, ours tackles at the level of systematics the ANN generalizability for non-frustrated systems.”

4) The following sentence added in page 4:

“see Appendix A.4 for a description of how the corresponding critical parameters are determined”

5) The following sentences added in page 9:

“An analysis of the ANN structures may reveal deeper insights about why such a minimal training strategy works, and this reduction may introduce simplification in the same or similar tasks in quantum machine learning. We plan to pursue these in a future work.”

---

## Editorial Decision

published